# END-TO-END VIDEO GENERATIVE MODELING WITH SCALABLE NORMALIZING FLOWS

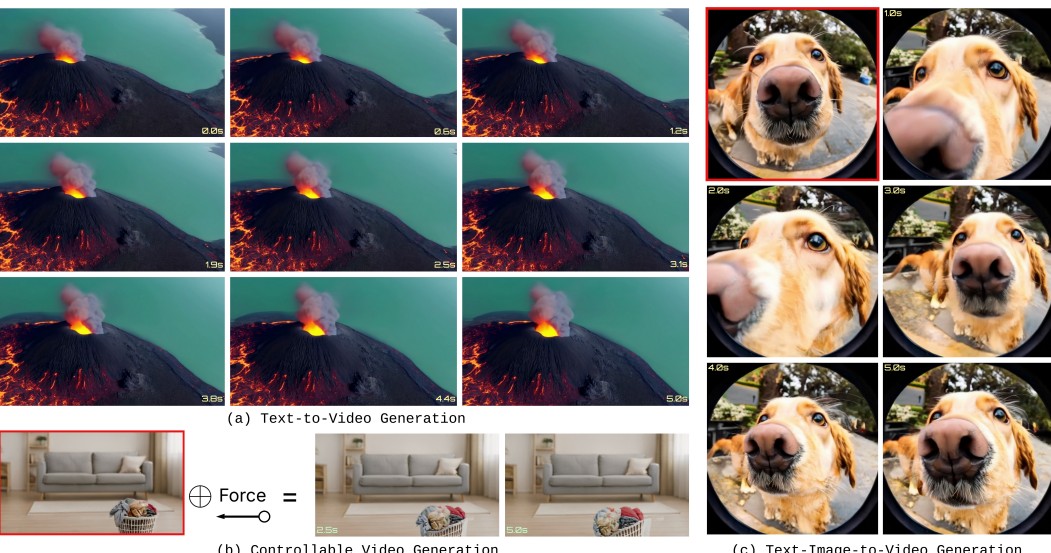

(a) Text-to-Video Generation

(b) Controllable Video Generation

(c) Text-Image-to-Video Generation

Figure 1: Samples from STARFlow-V in three tasks. All videos at 5 seconds with 16 FPS.

## ABSTRACT

High-quality video generation at scale requires models that are strictly causal, robust over long horizons, and fast at inference. We present STARFlow-V, a flow-based autoregressive video generator that operates in compressed spatiotemporal latents and is trained with exact likelihood end-to-end. Two design choices ensure causality for autoregressive prediction while mitigating error propagation and enabling end-to-end training: (i) Global–Local architecture, which constrains each token to depend only on the past along time while preserving rich within-frame interactions; and (ii) noise-augmented training jointly with *flow-score matching*, a lightweight causal denoiser that recovers clean samples from noisy generation. To improve efficiency, STARFlow-V employs a video-aware fixed-point iteration scheme that reformulates inner updates as parallelizable iterations without violating causal structure, yielding substantially faster inference. A deep–shallow autoregressive-flow hierarchy further balances capacity and stability over long videos. The same model natively supports both text-to-video (T2V) and text-/image-to-video (TI2V) generation via unified conditioning, avoiding separate pipelines. Empirically, STARFlow-V achieves strong visual fidelity and temporal consistency with markedly lower sampling cost compared to diffusion-only or discrete AR baselines. By marrying causality, likelihood, and efficiency in a single architecture, STARFlow-V helps pave the way toward a flow-based, scalable paradigm for world modeling.

# 1 INTRODUCTION

Generative modeling has advanced rapidly with breakthroughs across language (Achiam et al., 2023; OpenAI, 2024a), images (Podell et al., 2023; Batifol et al., 2025; Wu et al., 2025), and videos (OpenAI, 2024b; Wan et al., 2025; DeepMind, 2025). Among these modalities, *video* is uniquely demanding: beyond high perceptual quality, models must capture rich spatiotemporal structure, remain robust over long horizons, and often operate causally for interactive or streaming use. Such capabilities are central not only to creative media (Ye et al., 2025; Yuan et al., 2025), but also to emerging *world models* that support simulation, robotics, and human–AI interaction (Ha & Schmidhuber, 2018; Yang et al., 2023; Hu et al., 2023; Google DeepMind, 2024; Hafner et al., 2025).

Recent scaling of data, model capacity, and compute has pushed video generation to new levels of fidelity (Yang et al., 2025; Kong et al., 2024; Kondratyuk et al., 2024; Yu et al., 2024; Wan et al., 2025; Seawead et al., 2025; Gao et al., 2025). *Diffusion-based* approaches (Ho et al., 2020; Rombach et al., 2022; Peebles & Xie, 2023; Lipman et al., 2023; Esser et al., 2024) dominate quality by jointly denoising multiple frames; however, their training remains *not end-to-end*: frames are corrupted at randomly sampled noise levels and a denoiser is learned to invert them, so each update supervises only a single noise level, incurring high training cost (especially for video) and multi-step sampling at inference. Moreover, parallel multi-frame denoising is inherently *non-causal*, allowing future frames to influence earlier ones and complicating streaming or interactive generation. Recent sequential/causally conditioned diffusion variants (Chen et al., 2024a; Huang et al., 2025) alleviate non-causality via asynchronous noise schedules and post-training objectives, but they retain diffusion's training inefficiency and exhibit train–test mismatch during long-horizon rollout.

In this work, we revisit *normalizing flows* (Rezende & Mohamed, 2015; Dinh et al., 2014; 2016)—a family of invertible, likelihood-based generative models, distinct from diffusion approaches, that enable *end-to-end* training in continuous spaces—as a scalable foundation for video generation. Earlier attempts, such as VideoFlow (Kumar et al., 2019), were constrained by model capacity and the training practices of the time and saw no substantive follow-ups. In the image domain, recent systems (Zhai et al.; Gu et al., 2025) show that, by parameterizing the "autoregressive normalizing flow" with a Transformer, flows can scale competitively and approach diffusion-level quality.

Building on these insights, we present STARFlow-V, a normalizing-flow video generator operating on a spatiotemporal latent space. Specifically, STARFlow-V introduces **three** core contributions: (1) a *global–local* formulation that separates per-frame refinements from causal sequence modeling, preserving universality while easing error accumulation and enabling streamable generation; (2) noise-augmented training jointly with *flow-score matching*, a lightweight causal denoiser that recovers clean samples from noisy generation; and (3) efficient inference algorithm via video-aware blockwise Jacobi iteration and pipelined decoding that markedly reduces sampling latency. These advances make autoregressive flow inference tractable at scale while keeping training strictly end-to-end. As a result, STARFlow-V scales normalizing flows to high-fidelity, long-horizon video across various senarios including text-to-video, image-to-video, and controllable generation.

Extensive experiments demonstrate that STARFlow-V achieves competitive visual quality and robust generation compared to leading diffusion approaches,especially in autoregressive approaches. We believe STARFlow-V opens up a new direction in video generative modeling combining the scalability and expressivity of modern architectures with the principled advantages of end-to-end training in the continuous space.

# 2 BACKGROUND

## 2.1 VIDEO GENERATIVE MODELS

Given $N$ frames $\boldsymbol{x}_{1:N} = (\boldsymbol{x}_1, \ldots, \boldsymbol{x}_N)$ and optional conditioning $C$ (*e.g.*, text, image, audio, layout, camera), video generative models seek to model the joint distribution of all frames $p(\boldsymbol{x}_{1:N} \mid C)$ and sample novel videos from the learned model. While earlier work explored GANs (Vondrick et al., 2016; Tulyakov et al., 2018; Skorokhodov et al., 2022), VAEs (Babaeizadeh et al., 2018; Castrejon et al., 2019; Wu et al., 2021), and discrete autoregressive models (Yan et al., 2021; Yu et al., 2024; Kondratyuk et al., 2024), the field has largely converged on diffusion-based methods Ho et al. (2022c;a). Spurred by the release of Sora (Brooks et al., 2024), DiT-style approaches (Peebles & Xie,

2023) have shown strong generalization at scale (Gao et al., 2025; Wan et al., 2025; DeepMind, 2025). A key distinction from prior paradigms is that training of diffusion-based models is *Not End-to-End*: diffusion-based models corrupt frames with noise at randomly sampled levels and train a denoiser to invert this process, optimizing an objective closely related to the lower bound of $\log p(\boldsymbol{x}_{1:N} \mid C)$. This setup incurs high training cost—especially for video—since each update supervises only a single noise level. At inference time, one sample by iteratively denoising from Gaussian noise.

**Autoregressive Video Generation** Diffusion-based video generation is typically non-causal: all frames are corrupted with noise and denoised in parallel (Ho et al., 2022c). Yet many applications demand causal, often interactive synthesis (*e.g.*, online streaming, video games, robotics), where frames must be produced sequentially. Autoregressive (AR) diffusion models (Chen et al., 2024a; Song et al., 2025; Yin et al., 2025)—a line of work that combines chain-rule factorization with diffusion—aim to alleviate prior limitations by introducing asynchronous, frame-wise noise schedules during training, modeling each conditional $p(\boldsymbol{x}_n \mid \boldsymbol{x}_{<n})$ as a diffusion process. Despite their strengths, AR generation typically suffers from *exposure bias*: during training, models condition on ground-truth contexts, whereas at inference they must rely on their own (imperfect) predictions. This train–test mismatch compounds over time, degrading long-horizon video quality. The *non–end-to-end* nature of diffusion training further exacerbates this gap, though recent efforts such as Self-Forcing (Huang et al., 2025) seek to mitigate it via sequential post-training with distillation objectives. However, they are not readily applicable in the pre-training stage on raw video data.

**Video Latent Space** Directly modeling long-duration and high-resolution videos in pixel space is computationally challenging. Therefore, recent models typically operate in a compressed latent space (Rombach et al., 2022). In particular, video frames are encoded with a 3D causal variational autoencoder (VAE) (Yang et al., 2025; Wan et al., 2025), compressing both spatial and temporal dimensions while enforcing causality along the temporal axis. Throughout, we adopt latent-space representations unless explicitly indicated.

## 2.2 SCALABLE NORMALIZING FLOWS

Normalizing flows (NFs; Rezende & Mohamed, 2015; Dinh et al., 2014; 2016; Kingma & Dhariwal, 2018; Ho et al., 2019) are likelihood-based generative models built from invertible transformations. Given a continuous input $\boldsymbol{x} \sim p_{\text{data}}$, $\boldsymbol{x} \in \mathbb{R}^D$, an NF learns a bijection $f_\theta : \mathbb{R}^D \to \mathbb{R}^D$ that maps data $\boldsymbol{x}$ to latents $\boldsymbol{z} = f_\theta(\boldsymbol{x})$. Unlike diffusion models, NFs are trained *end-to-end* via a tractable maximum-likelihood objective derived from the change-of-variables formula:

$$\mathcal{L}_{\text{NF}}(\theta) = \mathbb{E}_{\boldsymbol{x} \sim p_{\text{data}}}[\log p_{\text{NF}}(\boldsymbol{x}; \theta)] = \mathbb{E}_{\boldsymbol{x} \sim p_{\text{data}}}\big[\log p_0\big(f_\theta(\boldsymbol{x})\big) + \log|\det(J_{f_\theta}(\boldsymbol{x}))|\big], \quad (1)$$

where the first term encourages mapping data to high-density regions of a simple prior $p_0$ (e.g., standard Gaussian), and the Jacobian term $J_f$ accounts for the local volume change induced by $f_\theta$, preventing collapse. Once trained, sampling is immediate via inversion: draw $\boldsymbol{z} \sim p_0(\boldsymbol{z})$ and set $\boldsymbol{x} = f_\theta^{-1}(\boldsymbol{z})$. Historically, however, NFs have been viewed as less competitive than diffusion models due to architectural rigidity and training instability (Dinh et al., 2016).

**Transformer Autoregressive Flows** Recently, TARFlow (Zhai et al.) and its scalable extension, STARFlow (Gu et al., 2025), have revisited normalizing flows as next-generation backbones for generative modeling. Both methods instantiate autoregressive flows (AFs) (Kingma et al., 2016; Papamakarios et al., 2017)—NFs whose invertible transformations are parameterized autoregressively—and use causal Transformer blocks, in the style of LLMs, as their primary building units.

Formally, STARFlow (Gu et al., 2025) stacks $T$ autoregressive flow blocks with alternating directions, where each block applies an affine transform whose parameters are predicted by a causal Transformer under a (self-exclusive) causal mask $\boldsymbol{m}$:

$$\boldsymbol{z} = f_\theta(\boldsymbol{x}) = \big[\boldsymbol{x} - \mu_\theta(\boldsymbol{x} \odot \boldsymbol{m})\big] / \sigma_\theta(\boldsymbol{x} \odot \boldsymbol{m}), \qquad \sigma_\theta(\cdot) > 0, \quad (2)$$

where $\boldsymbol{x}, \boldsymbol{z}$ are the input and output of each block, $\odot$ denotes the Hadamard product. As shown in STARFlow (Gu et al., 2025), $T \geq 3$ blocks suffice for universal density modeling where masks alternate between left-to-right ($\rightarrow$) and right-to-left ($\leftarrow$) to capture bidirectional dependencies.

**Video Generation with Normalizing Flows** Despite STARFlow demonstrating competitive visual quality with state-of-the-art diffusion (Podell et al., 2023; Esser et al., 2024) on large-scale text-to-image tasks, evidence for normalizing flows in video generation remains sparse. To our knowledge, the

only prior NF-based video model is VideoFlow (Kumar et al., 2019), which builds on Glow (Kingma & Dhariwal, 2018) and is constrained by limited capacity, low resolution, and domain-specific settings. Compared to images, video generation is substantially more challenging for NFs due to higher spatiotemporal dimensionality. Nevertheless, we argue that scalable normalizing flows—exemplified by STARFlow—are a natural fit for video modeling, especially in autoregressive settings.

## 3 STARFLOW-V

We propose STARFlow-V, a novel paradigm for video generation based on normalizing flows. While inspired by STARFlow (Gu et al., 2025), STARFlow-V is not a direct port to the video domain; it introduces architectural redesigns tailored to spatiotemporal data. In what follows, we present the architecture and its autoregressive formulation (§ 3.1), the training procedure (§ 3.2), the inference pipeline (§ 3.3), and applications enabled by our model (§ 3.4).

### 3.1 PROPOSED MODEL

For a video $\boldsymbol{x} \in \mathbb{R}^{N \times H \times W \times D}$, each frame $\boldsymbol{x}_n$ is flatten into $\mathbb{R}^{HW \times D}$, i.e., $\boldsymbol{x}_n = (\boldsymbol{x}_{n,1}, \ldots, \boldsymbol{x}_{n,HW})$. Concatenating across frames yields a sequence of total $NHW$ tokens. As in standard practice, we operate in a compressed latent space using a pretrained 3D causal VAE (Wan et al., 2025). STARFlow-V models the joint distribution $p_\theta(\boldsymbol{x})$ by an invertible mapping $f_\theta$ through autoregressive transformations (Eq. (2)). Following Gu et al. (2025), $f_\theta$ is decomposed into a *deep–shallow* architecture, $f_\theta = f_D \circ f_S$: a small stack of *shallow* flow blocks with alternating (left-to-right / right-to-left) masks maps the input to intermediate latents $\boldsymbol{u} = f_S(\boldsymbol{x})$, and a *deep* causal-Transformer flow $f_D$ then autoregressively maps $\boldsymbol{u}$ to the prior, producing $\boldsymbol{z} = f_D(\boldsymbol{u})$. By change of variables,

$$p_\theta(\boldsymbol{x}) = p_0(\boldsymbol{z}) \left| \det J_{f_D}(\boldsymbol{u}) \right| \left| \det J_{f_S}(\boldsymbol{x}) \right|, \tag{3}$$

where $J_f$ denotes the Jacobian of $f$ and $p_0$ is a simple prior (e.g., standard Gaussian). This design allocates most capacity to the deep block $f_D$ for semantics modeling while the shallow stack $f_S$ ensures universal approximation of continuous densities. For conditional generation, the context $C$ is prepended only in the deep block.

**Global-Local Architecture** A naïve implementation requires no change from the original STARFlow other than lengthening the input sequence by the number of frames. By default, we assume that the deep block $f_D$ follows a natural left-to-right order (causal across frames, raster order within each frame), while the shallow stack $f_S$ alternates directions, beginning with the reverse order. Although feasible, this setup yields a non-causal model—similar to standard diffusion-based video generators.

Observing that $f_D$ is inherently autoregressive and that $f_S$ mainly provides local refinements, we adapt the design into a *global–local* structure: $f_S$ is restricted to operate within each frame, while only $f_D$ propagates global video context in a causal manner. More specifically, Eq. (3) can be re-expressed as an autoregressive factorization over frames $\boldsymbol{x}_n$:

$$p_\theta(\boldsymbol{x}) = \prod_{n=1}^{N} p_\theta(\boldsymbol{x}_n \mid \boldsymbol{x}_{<n}) = \prod_{n=1}^{N} p_D(\boldsymbol{u}_n \mid \boldsymbol{u}_{<n}) \left| \det J_{f_S}(\boldsymbol{x}_n) \right|, \tag{4}$$

where $\boldsymbol{u}_n = f_S(\boldsymbol{x}_n)$ denotes the local latents for frame $\boldsymbol{x}_n$. Here, the deep block is itself an autoregressive flow, capturing both intra-frame raster ordering and inter-frame causal dependencies.

Formulating STARFlow-V in a *global–local* manner (Eq. (4)) yields several benefits:

(a) **Universality.** Eq. (4) preserves the universal approximation guarantee of STARFlow (Gu et al., 2025): the local stack $f_S$ still realizes per-pixel infinite Gaussian mixtures via alternating causal masks, so expressivity is not curtailed by restricting $f_S$ to within-frame contexts.

(b) **Robustness.** Intuitively, Eq. (4) can be viewed as a **continuous language model for videos**: the deep-flow term $p_D(\boldsymbol{u}_n \mid \boldsymbol{u}_{<n})$ acts as a *Gaussian next-token predictor* (cf. the affine form in Eq. (2)) in latent space, while the shallow flow supplies the Jacobian factor $\left| \det J_{f_S}(\boldsymbol{x}_n) \right|$, yielding a flexible density over $\boldsymbol{x}$. Compared to modeling $\boldsymbol{x}$ directly (arbitrarily multimodal), the latent $\boldsymbol{u}$ is unimodal at each step, easier to regress, and more tolerant to small prediction errors. Crucially, the sampling phase via $f_D^{-1}$ conditions on previously generated *latents* rather than

Figure 2: An illustrated pipeline of STARFlow-V which shows (1) the proposed global-local architecture; (2) joint training with the learnable denoiser with the proposed Flow-score Matching.

pixels, so data-space errors do not propagate forward, mitigating the compounding error typical of autoregressive diffusion. Unlike diffusion-style noise conditioning (Ho et al., 2022b; Chen et al., 2024a), which compromises information to gain robustness and introduces extra parameters, our mappings $\boldsymbol{u} \leftrightarrow \boldsymbol{x}$ are exactly invertible, avoiding information loss by construction.

(c) **End-to-End Training.** The overall model remains a valid flow. Consequently, all parameters are trained by exact maximum likelihood via the change-of-variables objective—no per-step denoising schedule or surrogate loss—simplifying optimization and reducing train–test mismatch.

(d) **Streamable Generation.** At inference time, $f_D^{-1}$ samples $\boldsymbol{u}_n$ causally (token-by-token, frame-by-frame), and $f_S^{-1}$ decodes each frame independently given $\boldsymbol{u}_n$. This process enables causal, and potentially interactive video synthesis since later frames cannot influence earlier ones.

## 3.2 REVISITING NOISE-AUGMENTED TRAINING FOR VIDEOS

As observed by Zhai et al. (2024), injecting *small* noise into the data is crucial for stabilizing NF training. Concretely, we learn STARFlow-V on a $\sigma$-smoothed density $q_\sigma(\tilde{\boldsymbol{x}}) = (p * \mathcal{N}(0, \sigma^2 I))(\tilde{\boldsymbol{x}})$. A side effect is that the model naturally generates slightly noisy samples, requiring a post-processing steps to recover the clean ones.

**Decoder Fine-tuning** We initially followed STARFlow (Gu et al., 2025) adopting their strategy of fine-tuning the VAE decoder to denoise noisy latents using a GAN objective (Rombach et al., 2022). However, our preliminary experiments suggest that this approach is not readily applicable to *3D causal* VAEs: under Gaussian-noised latent inputs, the decoder fails to maintain temporal consistency in the generated videos due to limited receptive fields.

**Score-based Denoising** Instead of decoder fine-tuning, TARFlow (Zhai et al., 2024) proposes to denoise using the *learned flow* itself via score-based updates. For a noisy sample $\tilde{\boldsymbol{x}} \sim q_\sigma$, the probability–flow ODE gives $\partial_\sigma \tilde{\boldsymbol{x}} = -\sigma \nabla_{\tilde{\boldsymbol{x}}} \log q_\sigma(\tilde{\boldsymbol{x}})$. So for sufficiently small $\sigma$, a single Euler step yields the Tweedie estimator:

$$\boldsymbol{x} \approx \tilde{\boldsymbol{x}} - \sigma \, \partial_\sigma \tilde{\boldsymbol{x}} = \tilde{\boldsymbol{x}} + \sigma^2 \nabla_{\tilde{\boldsymbol{x}}} \log q_\sigma(\tilde{\boldsymbol{x}}). \tag{5}$$

With normalizing flows, we replace $q_\sigma$ by the learned density $p_\theta$, and compute $\nabla_{\tilde{\boldsymbol{x}}} \log p_\theta(\tilde{\boldsymbol{x}})$ via automatic differentiation through the flow, which amounts to an additional forward–backward pass. However, this score-based denoising presents two issues:

(a) **Noisy gradients.** The learned density $p_\theta$ is imperfect; its score $\nabla_{\tilde{\boldsymbol{x}}} \log p_\theta(\tilde{\boldsymbol{x}})$ often contains high-frequency noise, which manifests as bright speckle artifacts—especially for large motions.

(b) **Non-causality of the score.** Even if $p_\theta$ is modeled causally, the score $\nabla_{\tilde{\boldsymbol{x}}} \log p_\theta(\tilde{\boldsymbol{x}})$ is, by definition, global: the gradient at time $n$ depends on likelihood terms involving future frames $m > n$. This violates strict causality and undermines the promised streamable generation.

**Flow-Score Matching** To address these issues, we introduce a neural denoiser $s_\phi$ trained alongside the flow $f_\theta$ to regress the model's score:

$$\mathcal{L}_{\text{denoise}}(\phi) = \mathbb{E}_{\boldsymbol{x}, \boldsymbol{\epsilon}} \big\| s_\phi(\tilde{\boldsymbol{x}}) - \sigma \nabla_{\tilde{\boldsymbol{x}}} \log p_\theta(\tilde{\boldsymbol{x}}) \big\|_2^2, \qquad \tilde{\boldsymbol{x}} = \boldsymbol{x} + \boldsymbol{\epsilon}, \ \boldsymbol{\epsilon} \sim \mathcal{N}(0, \sigma^2 I). \tag{6}$$

At inference, we replace the raw score in the update (cf. Eq. (5)) with the learned denoiser $s_\phi$. This *flow-score matching* (FSM) is simple yet effective. First, the smooth inductive bias of neural networks suppresses stochastic high-frequency artifacts in $\nabla_{\tilde{\boldsymbol{x}}} \log p_\theta$. Second, we can encode causality directly in $s_\phi$, re-ensuring streamable behavior. Concretely, we parameterize $s_\phi$ with a one–frame look-ahead while remaining globally causal (one-step latency)[1]. We approximate the score at step $n$ by

$$s_\phi(\tilde{\boldsymbol{x}}_{\leq n+1}) \approx \big(\sigma \nabla_{\tilde{\boldsymbol{x}}} \log p_\theta(\tilde{\boldsymbol{x}})\big)_n. \tag{7}$$

Finally, we train $s_\phi$ jointly with $f_\theta$ at **minimal overhead**: since $f_\theta$ is trained by maximizing $\log p_\theta$, we cache the input gradients from the same backward pass and reuse it (detached) as the target for $s_\phi$.

## 3.3 Fast Inference

While STARFlow-V leverages parallel computation during training via causal masking, generation at inference time is carried out sequentially (one token at a time) through multiple AF blocks, which can be *extremely* computationally demanding for long video sequences. For instance, generating a 5s 480p video under 16 fps using a pre-trained 3B parameter model requires over 30 minutes, which is far from real-time application. To enable fast inference, we introduce two strategies:

**Nonlinear Jacobi Iteration** Rather than sampling continuous tokens strictly autoregressively, we accelerate inference by recasting inversion as solving a nonlinear fixed-point system with parallel solvers such as Jacobi iteration (Porsching, 1969; Kelley, 1995), a strategy recently used to speed up autoregressive models (Song et al., 2021; Teng et al., 2024; Liu & Qin, 2025; Zhang et al., 2025). Specifically, the inverse of Eq. (2) can be written as the fixed-point equation

$$\boldsymbol{x} = \mu_\theta(\boldsymbol{x} \odot \boldsymbol{m}) + \sigma_\theta(\boldsymbol{x} \odot \boldsymbol{m}) \cdot \boldsymbol{z}, \tag{8}$$

where $\boldsymbol{m}$ is a (self-exclusive) causal mask. This induces a *triangular* system that admits convergence under nonlinear Jacobi iteration (Saad, 2003): starting from an initial sequence $\boldsymbol{x}^{(0)}$, iterate $\boldsymbol{x}^{(k+1)} = \mu_\theta(\boldsymbol{x}^{(k)} \odot \boldsymbol{m}) + \sigma_\theta(\boldsymbol{x}^{(k)} \odot \boldsymbol{m}) \cdot \boldsymbol{z}$ until a converge criterion is satisfied. We monitor a scale-normalized residual, $\|\boldsymbol{x}^{(k+1)} - \boldsymbol{x}^{(k)}\|_2^2 / \|\boldsymbol{x}^{(k+1)}\|_2^2 < \tau$ with $\tau = 0.001$ by default. In the worst case, the iteration count scales with sequence length (*e.g.*, near-Markovian process), but video generation exhibits strong global structure, which substantially accelerates convergence. The procedure is also *CFG-compatible*: following Gu et al. (2025), we compute the guided $\hat{\mu}, \hat{\sigma}$ and substitute them.

To further accelerate sampling, we adopt a block-based Jacobi scheme in the spirit of Song et al. (2021); Liu & Qin (2025). We partition the token sequence into contiguous blocks of size $B$ and process blocks sequentially. Within each block we run the Jacobi updates, while states from completed blocks are cached as context (*e.g.*, KV cache) for subsequent blocks—analogous to standard AR inference. We also apply a video-aware initialization: for a new frame, $\boldsymbol{x}_{n+1}^{(0)}$ is initialized from the previously converged frame $\boldsymbol{x}_n^{(k)}$. Overall, we adopt block-based iteration within each AF block, yielding $\approx 10\times$ lower inference latency relative to standard autoregressive decoding.

**Pipelined Decoding** As described in § 3.1, the global–local design applies standard global left-to-right autoregression in the deep block $f_D$, while the shallow blocks $f_S$ traverse each frame independently. This enables a pipelined schedule (analogous to pipeline parallelism (Huang et al., 2019)): $f_D$ runs continuously without waiting on $f_S$, and, in parallel, $f_S$ threads consume $f_D$'s outputs, immediately refine them, and then denoise. Because $f_D$ is typically the slowest stage, end-to-end latency is dominated by the deep block.

## 3.4 Versatility Across Tasks

STARFlow-V is a versatile framework that can be trained for diverse video generation tasks. By default, STARFlow-V is trained for text-to-video generation on large-scale text–video pairs. Without modifying the backbone, we support the following settings:

---

[1]Strictly causal ($\leq n$) fails as temporal *differences* are pivotal to determining the denoising direction.

(a) **Image-to-Video Generation.** We directly treat the first frame as observed conditioning. Owing to the invertiblity, *no separate encoder is required*: we encode the observed frame via the flow forward to initialize the KV cache; subsequent frames are then generated autoregressively.

(b) **Long-Horizon (Streaming) Generation.** Our model generates videos far longer than those seen during training via a sliding-window (chunk-to-chunk) schedule in the deep block. After producing a latent chunk $u$, we warm-start the next step by rebuilding the KV cache: we re-run $f_D$ on the last $\Delta$ latents (the overlap) and then continue autoregression to synthesize the next $N - \Delta$ latents. $f_S$ then process the latents per frame, enabling streaming output. To mitigate boundary mismatch, we randomly drop the first frame during training to simulate chunk restarts.

(c) **Controllable Video Generation.** For controllable synthesis (e.g., camera-aware generation using poses/intrinsics), we freeze the backbone and train a control module that injects control tokens into the autoregressive steps. This enables precise control without altering the core architecture.

## 4 EXPERIMENTAL SETUP

**Datasets.** We construct a diverse and high-quality collection of video datasets to train STARFlow-V. Specifically, we leverage the high-quality subset of Panda-70M (Chen et al., 2024b) mixed with an in-house stock video dataset, with a total number of 70M text-video pairs. For all videos, we keep their raw captions, and apply a video captioner (Wang et al., 2024a) to generate a longer description to cover the details. The ratio of training using raw and synthetic captions during training is $1 : 9$. Besides, following previous works (Lin et al., 2024), we additionally include image joint training with 400M text-image pairs.

**Evaluation.** We perform both quantitative and qualitative evaluations on STARFlow-V, and compare against baselines using VBench (Huang et al., 2024), which benchmarks text-to-video generation across 16 dimensions, including quality, semantics, temporal consistency, and spatial reasoning.

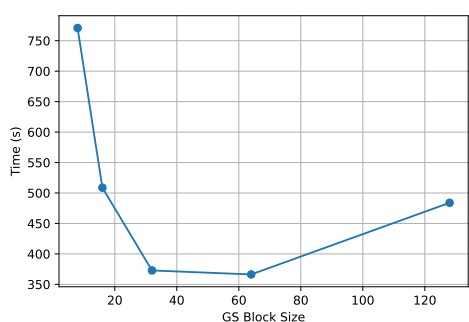

**Model and Training Details.** We adopt the 3D Causal VAE from WAN2.2[2] (Wan et al., 2025), which compresses spatial dimensions by $\times 16$ and the temporal dimension by $\times 4$ into a 48-channel latent space. We train progressively: we initialize from an image (single-frame) model, then scale to a 7B-parameter video model by increasing the deep-block capacity. For resolution,

Figure 3: Hyper-parameters of Parallel Iteration

we use a curriculum from 384p to 480p while keeping the sequence length fixed at 81 frames.

**Baselines.** We compare STARFlow-V with two baselines: (i) **WAN-2.1 Causal-FT**, the autoregressive variant of WAN (Wan et al., 2025) trained following the CausVid initialization strategy (Yin et al., 2025); and (ii) **NOVA** (Deng et al., 2024), an autoregressive video generator that does not rely on vector quantization.

## 5 RESULTS AND DISCUSSION

### 5.1 QUANTITATIVE RESULTS

Table 1 reports the text-to-video generation results on VBench (Huang et al., 2024). We show that STARFlow-V achieves *competitive* performance compared to diffusion-based methods.

---

[2] https://huggingface.co/Wan-AI/Wan2.2-TI2V-5B/blob/main/Wan2.2_VAE.pth

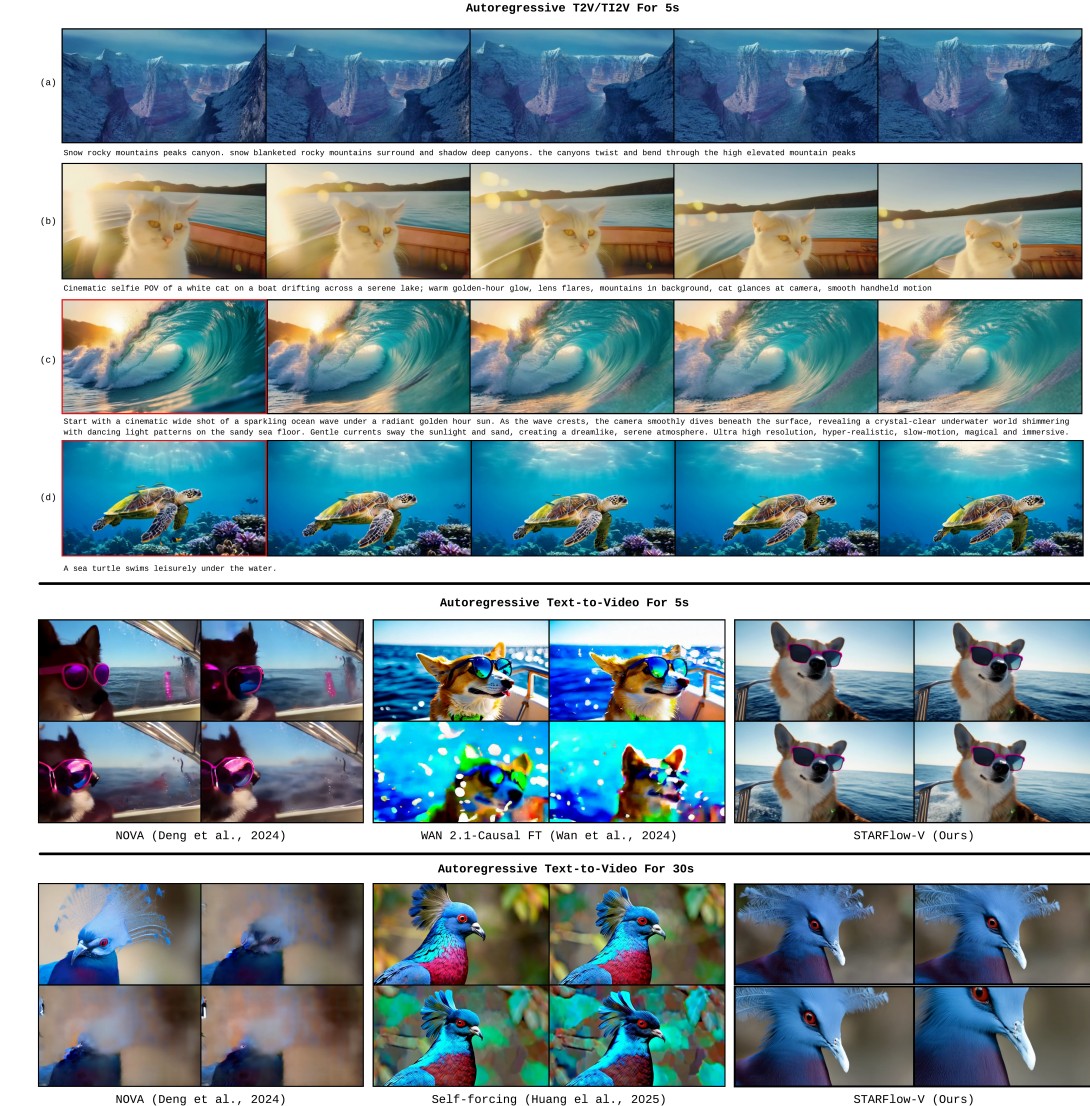

Figure 4: STARFlow-V examples of text and image conditioned video generation with comparison against baselines for both trained length (5s) and long-horizon generation (30s).

## 5.2 QUALITATIVE RESULTS

As illustrated in Fig. 4 (top block), STARFlow-V effectively handles both text-to-video (T2V) and image-to-video (I2V) generation. The first two rows show text-conditioned result and

Moreover, STARFlow-V produces consistent and high-quality videos even for videos extended to 30 seconds. Compared to other autoregressive video models, STARFlow-V demonstrates stronger robustness to exposure bias, which is a typical failure mode observed in autoregressive video generation, while retaining sharp textures and high visual quality over a long horizon.

In the dog-with-sunglasses example (Fig. 4, middle block), NOVA generates frames that blur and lose identity over time, while WAN 2.1-Causal FT suffers from severe artifacts and color distortions. By contrast, STARFlow-V produces clean, stable, and consistent frames across the sequence.

## 5.3 ABLATION STUDY

**Choice of Denoiser** Fig. 5 provides ablation evidence on the choice of denoiser, illustrated with two consecutive frames from three denoising strategies. Specifically, we show that Decoder-finetuning,

| Model | Total | Quality | Semantic | Aesthetic | Object | Multi Obj. | Human | Spatial | Scene |
|---|---|---|---|---|---|---|---|---|---|
| *Closed-source models* | | | | | | | | | |
| Gen-2 (Germanidis, 2023) | 80.58 | 82.47 | 73.03 | 66.96 | 90.92 | 55.47 | 89.20 | 66.91 | 48.91 |
| Gen-3 (Germanidis, 2024) | 82.32 | 84.11 | 75.17 | 63.34 | 87.81 | 53.64 | 96.40 | 65.09 | 54.57 |
| Veo3 (Google DeepMind, 2025) | 85.06 | 85.70 | 82.49 | 63.81 | 93.89 | 82.20 | 99.40 | 84.26 | 57.43 |
| *Diffusion models* | | | | | | | | | |
| OpenSora-v1.1 (Zheng et al., 2024) | 75.66 | 77.74 | 67.36 | 50.12 | 86.76 | 40.97 | 84.20 | 52.47 | 38.63 |
| OpenSora-v1.2 (Zheng et al., 2024) | 79.76 | 81.35 | 73.39 | 56.85 | 82.22 | 51.83 | 91.20 | 68.56 | 42.44 |
| CogVideoX (Yang et al., 2024) | 80.91 | 82.18 | 75.83 | 60.82 | 83.37 | 62.63 | 98.00 | 69.90 | 51.14 |
| HunyuanVideo (Kong et al., 2024) | 83.24 | 85.09 | 75.82 | 60.36 | 86.10 | 68.55 | 94.40 | 68.68 | 53.88 |
| Wan2.1-T2V (Wan et al., 2025) | 83.69 | 85.59 | 76.11 | 66.07 | 86.28 | 69.58 | 95.40 | 75.39 | 45.75 |
| *Autoregressive (Diffusion) models* | | | | | | | | | |
| CogVideo (Hong et al., 2022) | 67.01 | 72.06 | 46.83 | 38.18 | 73.40 | 18.11 | 78.20 | 18.24 | 28.24 |
| Emu3 (Wang et al., 2024b) | 80.96 | 84.09 | 68.43 | 59.64 | 86.17 | 44.64 | 77.71 | 68.73 | 37.11 |
| NOVA (Deng et al., 2024) | 80.12 | 80.39 | 79.05 | 59.42 | 92.00 | 77.52 | 95.20 | 77.52 | 54.06 |
| SkyReel-v2 (Chen et al., 2025) | 83.90 | 84.70 | 80.80 | - | - | - | - | - | - |
| MAGI-1-distill (Teng et al., 2025) | 77.92 | 80.98 | 65.68 | 62.43 | 82.37 | 35.08 | 84.20 | 57.75 | 26.28 |
| *Normalizing Flows* | | | | | | | | | |
| STARFlow-V (Ours) | 78.67 | 80.24 | 72.37 | 54.48 | 86.65 | 53.48 | 94.00 | 49.84 | 47.08 |
| STARFlow-V (Ours, with Rewriter) | 79.53 | 80.78 | 74.55 | 59.73 | 80.51 | 56.04 | 97.20 | 66.53 | 50.76 |

Table 1: **Text-to-video evaluation on VBench.** The baseline data is sourced from the VBench leaderboard (Huang et al., 2024).

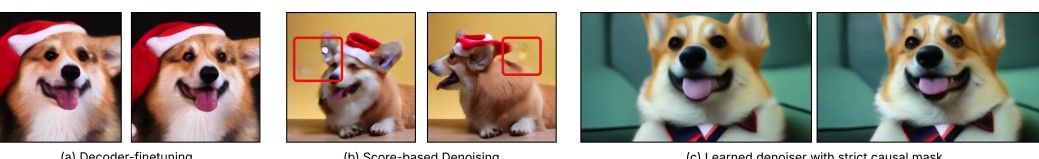

(a) Decoder-finetuning          (b) Score-based Denoising          (c) Learned denoiser with strict causal mask

Figure 5: Ablation Study for the choice of flow-score matching.

as in STARFlow (Gu et al., 2025), produces videos that lose temporal consistency, with evident frame-to-frame jitter. Score-based denoising, which uses the raw flow score, shows bright speckle artifacts—especially for large motions. In comparison, STARFlow-V shows temporally consistent and artifact-free videos.

**Hyper-parameters of Parallel Iteration** We analyze how the block size influences the runtime of the deep block. As shown in Fig. 3, larger group sizes increase parallelism but also introduce higher per-iteration overhead, while smaller groups reduce overhead but limit parallel efficiency. Our experiments reveal a favorable trade-off at moderate group sizes, which balances runtime efficiency with generation quality. In particular, a block size of 64 achieves the most favorable efficiency, and we adopt this setting for all experiments.

## 6 CONCLUSION AND LIMITATIONS

We presented STARFlow-V, an end-to-end video generative model based on normalizing flows. Across text-to-video and image-to-video, STARFlow-V delivers strong long-horizon coherence and fine-grained controllability, showing consistent gains over WAN-2.1 Causal-FT and NOVA at 480p/81f while providing exact likelihoods and streamable decoding via blockwise Jacobi and pipelined inference.

There are also limitations. (1) *Throughput/latency.* Despite the blockwise Jacobi acceleration and pipelining, inference remains far from real time on commodity GPUs. (2) *Data quality and scaling.* Progress is bounded by dataset noise and bias; we do not observe a clean scaling law under current curation, which constrains further improvements.

Looking forward, we plan to (i) reduce latency with kernel-level optimizations and partial-update decoders, (ii) study distillation and pruning to compress the deep block, and (iii) revisit dataset curation and active data selection to enable clearer scaling behavior and higher fidelity at longer durations and higher resolutions.

ETHIC STATEMENTS

**Ethic Considerations** :Our video generative model has the potential to enable new forms of creativity, data augmentation, and simulation. However, it also raises important ethical concerns. In particular, the ability to generate realistic video content carries risks of misuse, including the creation of misleading or harmful media. Such risks highlight the importance of establishing safeguards around model deployment and access.

**The use of Large Language Model (LLM)** A large language model (LLM) was employed solely for stylistic polishing of the manuscript. It was not used for generating scientific content, conducting analyses, or contributing to the conceptual development of this work. All technical ideas, methods, and results are entirely the author's own.

REPRODUCIBILITY STATEMENTS

We are committed to ensuring the reproducibility of our work. Upon acceptance, we will release the complete codebase, including all training details, hyperparameters, and model configurations. For privacy reasons, the raw data used in our experiments cannot be released. Theoretically, we confirm that all variables used in the equations are well-defined to facilitate the reproducibility of our work. Practically, to further enhance understanding, we provide extensive visual illustrations (e.g., Fig. 2) to support reproducibility.

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
