# OpenReview forum: "End-to-End Video Generative Modeling with Scalable Normalizing Flows"
_ICLR.cc/2026/Conference — ICLR 2026 Conference Withdrawn Submission_

### Official Review · Reviewer_jjFR · 2025-10-18

**Soundness:** 1
**Presentation:** 2
**Contribution:** 2
**Rating:** 2
**Confidence:** 3

**Summary:**

This paper proposes a new video generation model based on STARFlow [Gu'2025], a type of normalizing flow. To address the computational cost and error propagation issues, the authors propose three key designs: (1) a global-local architecture where deep blocks model inter-frame global information, while shallow blocks focus on intra-fame local dependencies. (2) a noise-augmented training strategy that is jointly trained with a flow matching based denoiser (3) an efficient inference algorithm using blockwise parallel Jacobi iteration. The proposed method is evaluated on both text-to-video and text-image-to-video tasks, achieving comparable performance to diffusion-based video generation models.

**Strengths:**

1. The proposed three components (global-local architecture, noise-augmented training, blockwise inference) are well motivated and effectively tackle the main challenges in video generative modeling.
2. Experimental results on VBench demonstrate that the proposed normalizing flow-based method achieves comparable results to diffusion and autoregressive baselines.

**Weaknesses:**

1. Many claims remain unsupported.

   1. In Section 3.4, the authors claim their model is versatile and supports I2V, long-horizon generation, and controllable video generation. However, no experimental results are provided for controllable video generation. Furthermore, it is also unclear which examples correspond to I2V, as the relevant paragraph (lines 419–420) appears incomplete.
   2. The authors also assert "diffusion's training inefficiency" (line 74) and their "efficient inference algorithm" (line 87), yet offers insufficient evidence to support these claims. In particular, comparisons with baseline models in terms of training cost and inference speed are missing.It would be better for the authors to conduct additional quantitative analysis to support these claims.

  2. Lack of ablation studies. Although three designs are proposed, the paper does not include any ablation studies to verify the effectiveness of each design. It would be better if the authors could perform additional ablation study isolating the performance contribution of each component.
  3. Weak qualitative results. The qualitative comparisons in Figure 4 (bottom two rows) show that their generated videos exhibit limited motion, falling short of the motion dynamics achieved by current diffusion and autoregressive models. It would be better if the authors include a motion analysis (e.g.  based on VBench metrics), and conduct additional user study to justify their advantages.

**Questions:**

1. What is the model size of the proposed STARFlow-V, and what are the basic training details (e.g. number of training epoch, data filtering procedure)?

---

### Official Review · Reviewer_ASDa · 2025-10-30

**Soundness:** 3
**Presentation:** 3
**Contribution:** 3
**Rating:** 6
**Confidence:** 3

**Summary:**

STARFlow-V is an end-to-end, strictly causal video generator based on normalizing flows, extending STARFlow from images to videos. It operates in a compressed spatiotemporal latent space via a 3D causal VAE and is trained with exact maximum likelihood. The core Global–Local design uses a deep–shallow autoregressive-flow hierarchy: a shallow, frame-local invertible stack refines within-frame details and contributes the Jacobian term, while a deep autoregressive flow performs Gaussian next-token prediction over frame latents causally across time. This mitigates pixel-space error propagation and enables streamable generation.

Training stability and output cleanliness are addressed via noise-augmented learning and Flow-Score Matching: a lightweight, near-causal denoiser learns the model score with a one-frame look-ahead and applies a single Tweedie update at inference, avoiding non-causal gradients and artifacts. Efficiency is improved by casting inversion as a nonlinear fixed-point problem, enabling blockwise Jacobi iteration with video-aware initialization and pipelined decoding. A single backbone supports T2V, I2V, and controllable generation, achieving competitive VBench performance and strong long-horizon consistency.

**Strengths:**

1. A strictly causal, flow-based video generator with a Global–Local design and deep–shallow AF hierarchy. The near-causal Flow-Score Matching denoiser (one-frame look-ahead) enables single-step Tweedie cleanup without non-causal gradients. Efficient inference via fixed-point (blockwise Jacobi), video-aware init, and pipelined decoding reduces latency while preserving causality.

2. Broad coverage: 480p/81f T2V/I2V, streaming up to 30s, and control. Better than several AR diffusion baselines with lower sampling cost. ~10× latency reduction and CFG-compatible streaming show practical viability.

**Weaknesses:**

1. Claims of reduced exposure bias and improved temporal stability rely heavily on qualitative demos and VBench. There is no dedicated metric suite or failure-mode analysis for >30s sequences.

2. The paper reports ~10× speedup but does not deeply analyze its sources or limits. There is no component-wise attribution, nor end-to-end profiling across resolution, clip length and block size. Crucially, standardized compute and resource metrics are missing, e.g., FLOPs per frame/video, parameter count, throughput. Without these, claims of “scalable” and "efficient" are hard to reproduce, compare, or operationalize. The paper would benefit from detailed complexity tables, comprehensive hyperparameter sweeps, and component-level ablations that quantify individual and combined contributions to speed and quality.

**Questions:**

1. It would be much better if the paper provided video demos for side-by-side comparisons, including long-horizon and fast-motion cases, to make temporal coherence and artifact differences clearer and more convincing.

---

### Official Review · Reviewer_PJNu · 2025-10-30

**Soundness:** 2
**Presentation:** 2
**Contribution:** 2
**Rating:** 4
**Confidence:** 3

**Summary:**

This paper presents STARFlow-V, an autoregressive video generator that uses normalizing flows. Their method operates in a compressed latent space using a pretrained 3D causal VAE. They present competitive results on Vbench, although they still lag behind diffusion-based state-of-the-art like HunyuanVideo and Wan2.1-T2V.

**Strengths:**

1. Their method for video generation allows for end-to-end training, which is potentially more scalable in the future.
2. Their method performs an autoregressive generation of frames. In the future, this would allow for potential strategies to perform interventions/counterfactuals, allowing for more fine-grained interactive video generation. Their teaser figure (Fig 1b) where they apply a force on an object is similar to "World Modeling with Probabilistic Structure Integration" which introduces a visual world model that also does autoregressive generation and allows for hypotheticals/counterfactuals.

**Weaknesses:**

1. Their results on Vbench are slightly worse than the best diffusion-based models, e.g., HunyuanVideo and Wan2.1-T2V.
2. The teaser figure (Fig. 1) is presented with no explanation and not referenced in the paper, to my knowledge. In particular, I think they actually highlight limitations of the model and I also had some questions. (Figure 1a) It seems like there is almost no motion at all, so this makes the "video generation" capabilities of the model seem unimpressive. What was the text prompt used? (Figure 1b) How did you apply the force conditioning? The basket is a rigid object, the generated video seems physically implausible because the basket deforms. And why does it continue to deform from 2.5 to 5s, is the force applied throughout? (Figure 1c) What is the text prompt used?
3. For many of the generated videos (Fig. 4), there is pretty much no object motion, and very little camera motion. Also, for the wave crest video, the wave seems to be going upwards instead of falling, which seems physically implausible.

**Questions:**

1. How long does it take for inference, after implementing your two speedup strategies? How does this compare to methods that you compare performance to in your paper (e.g., HunyuanVideo and Wan2.1-T2V)?

---

### Official Review · Reviewer_Bm21 · 2025-11-03

**Soundness:** 3
**Presentation:** 3
**Contribution:** 2
**Rating:** 2
**Confidence:** 4

**Summary:**

This paper proposes STARFlow-V, a video generation framework based on normalizing flow that enables end-to-end training and streaming inference for video frames. Building upon STARFlow for image generation, STARFlow-V introduces (1) a Global-Local architecture for spatio-temporal modeling, (2) flow-score matching for stable and causal denoising, and (3) an efficient inference scheme combining blockwise Jacobi iteration with pipelined decoding. Initialized from an image model, STARFlow-V scales up to a 7B-parameter video model and is trained to generate 480p, 5s videos from text, text-and-image, or camera-control inputs in an autoregressive manner.

**Strengths:**

S1. Novel framework for end-to-end likelihood training of video generative modeling. In addition to the advance in STARFlow of image generation, STARFlow-V successfully extended the NF framework into video domain.

S2. The proposed sampling methods improve inference efficiency of normalizing flow based generative models for videos. Specifically, compared with the baseline, the proposed blockwise Jacobi iteration with pipelined decoding achieves 10x faster inference speed.

S3. The proposed method shows that an NF-based framework can also have a potential to train a billion scales of generative models for video tasks.

**Weaknesses:**

Although I believe the potential of this study to improve normalizing flow based generative models to be scalable, I have some concerns in the current version.

W1. Experiments are insufficient, while the results are not competitive yet. Although the proposed methods have theoretical background and make sense, the experimental results could not support its competitiveness yet. Unlike Line 92, extensive experiments are not conducted – only the VBench score of the final model and few qualitative comparisons are provided. In addition, the results are not competitive with diffusion-based methods because there are large gaps in many categories of VBench. For the performance of autoregressive decoding, Drifting metric [NewRef-1] can also be used. In addition, more thorough analysis of ablation study and competitor comparisons should be added because the current version only includes few qualitative comparisons.

W2. Scalability of this framework is not demonstrated. Although Lines 94-96 claim that this framework can be a new direction of video generative models for scalability, there is no supporting result in scalability. Rather, the sampling speed is still too slow (~6 minutes for 480p, 5s videos). In addition, contradictorily, the authors mentioned that this framework is not scalable yet. Considering this framework aims to train large-scale video generative models, proving the scalability is necessary to claim the efficacy of this framework in the current version.


[NewRef-1] Frame Context Packing and Drift Prevention in Next-Frame-Prediction Video Diffusion Models (Zhang et. al., 2025)

**Questions:**

Please refer to my comments on experiments and scalability above. In addition to the weakness, I have few questions to elaborate the contribution and the current status of this paper.


Q1. How can you claim the advantage of normalizing flow based large-scale models on end-to-end training? Is this framework much more efficient and effective than diffusion/flow-based models?

Q2. Why is the scaling behavior not shown in this paper? Can you provide the experimental results on scaling law and some analysis that the training data has some issues on scaling law? On the other hand, how can you claim that the issue comes from the current mix of training dataset not from the proposed framework or formulation itself?

Q3. I think Figure 4 shows more static results than the others. Can you provide a Dynamic score based on the RAFT model in [NewRef-1]? You can also add diverse metrics [NewRef-1] for thorough analysis of experimental results.

---

### Note · Authors · 2025-11-14

**Comment:**

We have decided to withdraw the paper because the current version was prepared under significant time constraints and lacks several key experiments. We sincerely thank all the reviewers for their constructive feedback, which we will carefully incorporate into a more complete resubmission!

**Withdrawal Confirmation:**

I have read and agree with the venue's withdrawal policy on behalf of myself and my co-authors.